# DWI Metrics Differentiating Benign Intraductal Papillary Mucinous Neoplasms from Invasive Pancreatic Cancer: A Study in GEM Models

**DOI:** 10.3390/cancers14164017

**Published:** 2022-08-20

**Authors:** Miguel Romanello Joaquim, Emma E. Furth, Yong Fan, Hee Kwon Song, Stephen Pickup, Jianbo Cao, Hoon Choi, Mamta Gupta, Quy Cao, Russell Shinohara, Deirdre McMenamin, Cynthia Clendenin, Thomas B. Karasic, Jeffrey Duda, James C. Gee, Peter J. O’Dwyer, Mark A. Rosen, Rong Zhou

**Affiliations:** 1Department of Radiology, University of Pennsylvania, Philadelphia, PA 19104, USA; 2Pancreatic Cancer Research Center, University of Pennsylvania, Philadelphia, PA 19104, USA; 3Abramson Cancer Center, University of Pennsylvania, Philadelphia, PA 19104, USA; 4Department of Pathology and Laboratory Medicine, University of Pennsylvania, Philadelphia, PA 19104, USA; 5Department of Biostatistics, Epidemiology and Informatics, University of Pennsylvania, Philadelphia, PA 19104, USA

**Keywords:** pancreatic adenocarcinoma, intraductal papillary mucinous neoplasms, neoplastic progression, diffusion-weighted MRI, dynamic contrast-enhanced MRI, genetically engineered mouse model

## Abstract

**Simple Summary:**

Intraductal papillary mucinous neoplasms (IPMN) are common premalignant precursors to pancreatic ductal adenocarcinoma (PDAC). However, current guidelines for clinical management of IPMN are suboptimal and limited to information obtained through anatomical imaging and cytology of biopsy samples. Our results have suggested that diffusion-weighted MRI, an imaging method sensitive to tumor cell density, architecture, and microenvironment is able to differentiate mouse models of IPMN versus PDAC.

**Abstract:**

KPC (Kras^G12D^:Trp53^R172H^:Pdx1-Cre) and CKS (Kras^G12D^:Smad4^L/L^:Ptf1a-Cre) mice are genetically engineered mouse (GEM) models that capture features of human pancreatic ductal adenocarcinoma (PDAC) and intraductal papillary mucinous neoplasms (IPMN), respectively. We compared these autochthonous tumors using quantitative imaging metrics from diffusion-weighted MRI (DW-MRI) and dynamic contrast enhanced (DCE)-MRI in reference to quantitative histological metrics including cell density, fibrosis, and microvasculature density. Our results revealed distinct DW-MRI metrics between the KPC vs. CKS model (mimicking human PDAC vs. IPMN lesion): the apparent diffusion coefficient (ADC) of CKS tumors is significantly higher than that of KPC, with little overlap (mean ± SD 2.24±0.2 vs. 1.66±0.2, $p<10^{-10}$) despite intratumor and intertumor variability. Kurtosis index (KI) is also distinctively separated in the two models. DW imaging metrics are consistent with growth pattern, cell density, and the cystic nature of the CKS tumors. Coregistration of ex vivo ADC maps with H&E-stained sections allowed for regional comparison and showed a correlation between local cell density and ADC value. In conclusion, studies in GEM models demonstrate the potential utility of diffusion-weighted MRI metrics for distinguishing pancreatic cancer from benign pancreatic cysts such as IPMN.

## 1. Introduction

Human pancreatic ductal adenocarcinoma (PDAC) has prevailing genetic signatures including KRAS^G12D^ mutation, Smad4 deletion, and/or TP53 mutation. Therefore, genetically engineered mouse (GEM) models provide powerful tools to study genetic drivers of tumor initiation and malignant transformation, impact of stromal cells, and novel interventions. Mice with KRAS^G12D^ mutation in the pancreatic epithelium develop premalignant pancreatic intraepithelial neoplastic (PanIN) lesions, which slowly progress to malignancy, i.e., PDAC. However, in the presence of simultaneous KRAS and TP53 mutations (Kras^G12D^:Trp53^R172H^:Pdx1-Cre), the progression is remarkably accelerated [1]: this GEM model, referred to as KPC, exhibits high penetrance and reproducible kinetics of tumors that resemble key characteristics of conventional human PDAC [2,3]. Recent development of GEM models revealed that adding Smad4 deletion to KRAS^G12D^ mutation (Kras^G12D^:Smad4^L/L^:Ptf1a-Cre), both of which target the pancreatic epithelium, leads to tumors that resemble human intraductal papillary mucinous neoplasms (IPMN) [4]; this GEM model is referred to as CKS. IPMN is considered a precursor to intraductal papillary mucinous carcinoma (IPMC), a subtype of pancreatic adenocarcinoma, and clinical management of this disease is suboptimal and is still being debated [5]; surgery is recommended based on certain radiographic features such as nodule size, location relative to and communication to the main pancreatic duct, and/or main pancreatic duct dilation [6]. However, the anatomical information obtained from standard imaging and cytological analysis of biopsy samples often do not provide sufficient information about the malignant transformation of the IPMN lesion. Ideally, a clinical imaging marker that separates benign IPMN from those becoming invasive carcinoma would inform the decision between continuing surveillance versus surgical resection. Diffusion-weighted (DW)-MRI and dynamic contrast enhanced (DCE)-MRI are two clinical in vivo imaging methods that can provide quantitative metrics sensitive to the tumor microenvironment and cellularity (density of cancerous cells within the tumor) [7]. We hypothesize that differences in cellular architecture and stromal features between PDAC and IPMN can be revealed by DW-MRI and/or DCE-MRI metrics. Although histopathological features have been previously characterized for both the CKS and KPC models, their impact on quantitative DW- or DCE-MRI metrics has not been examined and therefore is a central focus of this study. IPMC, the PDAC that arises from IPMN, may have different morphologic phenotypes as with usual pancreatic adenocarcinoma and may be histopathologically characterized into subtypes; the main driver of prognosis is the stage of extent of invasion. As the more prevalent and lethal subtype of IPMC presents similar histopathological features to conventional PDAC [8], the KPC model provides an adequate GEM model for the more invasive adenocarcinoma arising from IPMN, although it may not capture the full spectrum of invasiveness in a progressing IPMN.

For abdominal cancers such as liver and pancreatic cancer, respiratory motion is a major challenge in quantitative DCE- and DW-MRI because it causes distortions and blurring artifacts in the images [9,10,11], introducing errors to the DCE- and DW-MRI derived metrics. Based on our recent demonstrations that radial k-space sampling both mitigates respiratory motion artefacts [12,13] and enables improved spatial and temporal resolutions in DCE-MRI through k-space weighted image contrast (KWIC) reconstruction [14], we applied the motion-robust DW and DCE protocols to the two GEM models for this study. We aim to achieve the following goals: (1) to compare quantitative DW- and DCE-MRI metrics of KPC and CKS tumors; (2) to identify histological features that may corroborate these differences including cellular architecture, growth pattern, and quantitative histological measures such as cell density, fibrosis, microvasculature density, and necrosis; and (3) to demonstrate an approach for direct comparison of apparent diffusion coefficient (ADC) maps obtained from DW-MRI and cell density maps obtained from hematoxylin and eosin (H&E) staining. Our results reveal that DW-MRI metrics are distinct between KPC and CKS models with almost no overlap and are corroborated with IHC metrics. In addition, our data reveal a paradoxical finding between DCE-MRI metrics and tumor microvascular density.

## 2. Materials and Methods

All animal handling procedures were reviewed and approved by the Institutional Animal Care and Use Committee of the University of Pennsylvania. The number of mice enrolled in DW and DCE imaging and histopathological studies are indicated in figures and tables in Results section.

### 2.1. GEM Models of Pancreatic Cancer

KPC mice (Kras^G12D^:Trp53^R172H^:Pdx1-Cre) were bred in the Mouse Hospital, Abramson Cancer Center of the University of Pennsylvania. CKS mice (Kras^G12D^:Smad4^L/L^:Ptf1a-Cre) were bred in the Small Animal Imaging Facility of the University of Pennsylvania. KPC mice develop premalignant PanIN at 7–10 weeks of age, leading to invasive PDAC at 17–19 weeks. Screening for tumors was done via weekly abdominal palpation starting at 11 weeks of age followed by ultrasound examination (Vevo 2100, VisualSonics, Toronto, ON, Canada) to estimate tumor size. Mice with a confirmed tumor mass (both sexes, 18–25 weeks old) were enrolled in the MRI study. Based on a prior report [4] and our data, CKS mice develop normally including normal pancreas and pancreatic function. Over time, predominantly in the large pancreatic ducts, in a multifocal manner, neoplastic mucinous epithelial proliferation ensues with papillary enfolding and intraductal growth. Ducts are continuously expanded by the intraductal papillary growth, which compresses the adjacent pancreatic tissue. CKS mice present palpable tumor mass at 7–12 weeks of age and have rapid tumor progression afterwards [4]. Both KPC and CKS mice were enrolled in DW- and DCE-MRI protocols after a tumor mass was confirmed by MRI.

### 2.2. In Vivo MRI, Image Reconstruction and Analyses

In vivo MRI was performed on a 9.4 T Avance III console (Bruker, Berillica, MA, USA) equipped with a 12 cm, 40 gauss/cm gradient coil with a maximum slew rate of 11.5 T/cm·s. The mice were induced by 3% isoflurane mixed in air and remained under anesthesia during MRI by breathing 1–3% isoflurane air mixture via a nose cone. The mice were positioned prone in a custom bed with slots for two sealed 5 mm NMR tubes (Wilmad Glass, Vineland, NJ, USA) containing 10% and 40% (*w*/*w*) polyvinylpyrrolidone (PVP, Sigma-Aldrich, St. Louis, MO, USA) aqueous solutions. A pneumatic pillow was placed underneath the abdomen and a lubricated temperature sensor inserted in the rectum (both purchased from SAI Inc., Stonybrook, NY, USA). The bed was then mounted into a 35 × 40 mm (ID × length) quadrature birdcage RF coil (M2M, Cleveland, OH, USA). During MRI exams, the core temperature was maintained at 37 ± 0.2 °C by directing a temperature regulated air source into the bore, while the respiration rate was maintained in the range of 80–100 breaths/min by adjustment of the isoflurane level.

Following power calibration and generation of scout images, axially oriented contiguous $T_{2}$ weighted (T2W) images spanning the entire tumor were generated using a 2D TurboRARE protocol (effective TE = 30 ms, TR = 1.8 s, echoes = 8, matrix = 128 × 128, slices = 16, field of view (FOV) = 32 × 32 mm^2^, thickness = 1.5 mm, averages = 4, total acquisition time ∼2 min); this T2W series served as reference for planning the DW- and DCE-MRI scans and for identifying the PDAC tumors.

#### 2.2.1. In Vivo DWI Acquisition, Image Reconstruction and Analyses

Acquisition protocol was described earlier [12] and optimized with the following parameters: a radial k-space sampled diffusion-weighted spin echo sequence was applied with diffusion time (Δ) = 14 ms, diffusion gradient duration (δ) = 9 ms, *b* = 10, 535, 1070, 1479, and 2141 mm^2^/s, TE/TR = 28.6/750 ms, FOV = 32 × 32 mm^2^, matrix size = 96 × 403, slices = 16, thickness = 1.5 mm, averages = 1, bandwidth = 50 kHz, total acquisition time ∼25 min. No respiration gating was employed. Fat saturation pulses were applied.

For data reconstruction, zero order phase correction was applied to each radial spoke and the resulting data was regridded [15] to a Cartesian matrix (96 × 96). To obtain parametric maps, pixel intensities in DW images were fit to a three-parameter exponential decay model (Equation (Equation 1)) using the least squares method, where S(b) is image intensity as a function of b-values, M_0_ is the intensity without diffusion weighting, and BL is the baseline signal as the exponential component in Equation (Equation 1) approaches to zero at high b-values. The reconstruction and least squares fitting were carried out using custom Python code to generate pixelwise parametric maps of ADC, M_0_ and kurtosis index (KI), defined by Equation (Equation 2). In previous studies conducted on phantoms over a large range of b-values, we found that this definition accurately modeled kurtosis.
(1)$$S\left(b\right)=M_{0}e^{-b*ADC}+BL,$$
(2)$$\mathrm{Kurtosis}\;\mathrm{Index},\;KI=BL/M_{0}.$$

#### 2.2.2. In Vivo DCE-MRI Acquisition, Image Reconstruction and Analyses

A 27-gauge catheter was inserted into a lateral tail vein of the mouse. This catheter was then connected to a 6 ft long PE-10 tube prefilled with contrast media solution containing 10 mM Gd by dilution of ProHance (Bracco, NJ, USA) with PBS. The DCE protocol consisted of three components: (1) Actual Flip-Angle Imaging (AFI) for correction of radio frequency field inhomogeneity (B1); (2) mapping of the longitudinal relaxation time ($T_{1}$); (3) the DCE series acquired continuously before, during, and after contrast media injection. All three components employ the stack-of-stars (SoS) sequence, a 3D hybrid radial k-space sampling method comprised of multiple 2D radial GRE encoding in-plane and phase-encoding in the slice direction. The AFI protocol was applied with a pair of outer volume suppression pulses to reduce the high signal from inflowing unsaturated blood in the great vessels (TE/TR1/TR2 = 1.2/13/65 ms, flip angle = 60°, averages = 2, total acquisition time ∼8 min). $T_{1}$ mapping employs the variable flip angle (VFA) method with spoiled GRE protocol (TR/TE = 5.5/1.25 ms, FA = 2, 5, 8, 12, 16, 20°, averages = 4, total acquisition time ∼7 min). Lastly, the DCE series was acquired according to the golden-angle ordering scheme, where the angle of the radial views is increased continuously by 111.25° to obtain a uniform coverage of k-space (TR/TE = 5.5/1.25 ms, flip angle = 9°, averages = 1, repetitions = 40, total acquisition time ∼12 min). At 2 min after starting of DCE series, 0.2 mL Gd solution was injected over 10 s into the catheter tubing manually. All three components of the DCE protocol (AFI, VFA, DCE series) had matrix size = 128 × 201 × 16 and FOV = 32 × 32 × 24 mm^3^.

For image reconstruction of data generated by the AFI and $T_{1}$ mapping protocols, after Fourier transformation in the slice dimension, radial k-space data were phase corrected and regridded to a 128 × 128 Cartesian array. The regridded data were then Fourier transformed in the remaining two dimensions. For the DCE series, the k-space weighted image contrast (KWIC) method described by Song et al. [16,17] was employed to reconstruct the images to 128 × 128 Cartesian space with 3 levels, 50 views encoding the k-space center region and a sliding window of 50 views, resulting in 157 temporal frames per slice location with an effective temporal resolution of 4.4 s.

B1 field maps were generated from the AFI images using the method described by Yarnykh et al. [18]. The flip angle calculated via the two AFI images was divided by the nominal flip angle used in the sequence (60°) to yield the normalized field maps. The normalized field maps were used for B1 correction of $T_{1}$ maps and during reference region model analysis of the DCE series, as described in detail recently [13]. $T_{1}$ maps were generated by a nonlinear least squares fit to the Ernst equation.

For kinetic modeling of the DCE series, a reference region model (RRM) using muscle as the reference tissue [19,20] was employed. The RRM was coded in custom Python code to derive the DCE-MRI metrics including the transfer constant of contrast agent from capillaries to interstitial space ($K^{trans}$ in /min) and extracellular/extravascular volume fraction ($v_{e}$). $K^{trans}$ and $v_{e}$ of the reference tissue, the spinal muscle, were assumed to be 0.10/min and 0.10, respectively [19], and 4.6/s·mM was used as the relaxivity of the contrast media.

#### 2.2.3. ROI Drawing and Analyses

For both DW- and DCE-MRI, regions of interest (ROIs) were manually drawn for phantoms, tumor, and spinal muscle in all slices in which they appeared (Appendix A) using open-source program ImageJ (https://imagej.nih.gov/ij/, version 1.53k, accessed on 16 August 2021). Due to potential slight position shift of the mouse between the DWI and DCE scans, ROIs were drawn separately for each image. Pixels in ROIs of a specific tissue or phantom were then combined to derive median and standard deviation (SD) of MRI metrics (ADC, KI, $T_{1}$, $K^{trans}$ and $v_{e}$) for each subject.

### 2.3. Ex Vivo DWI, Coregistration of Ex Vivo ADC Map with H&E-Stained Section and Correlation of ADC with Cell Density

1 mm thick sections were cut from freshly harvested tumor specimens of five KPC mice. Each section was placed in a custom tissue vial (ID = 3 cm) filled with perfluoropolyether (Fomblin^®^ Solvay Solexis, Thorofare, NJ, USA). The vial was loaded into a custom 11 mm ID high resolution RF coil constructed in-house, which was inserted into 8.9 cm ID vertical bore 9.4 T NMR spectrometer (DirectDrive^®^, Agilent) equipped with a 55 mm ID gradient coil with a maximum strength of 100 gauss/cm. DW images were acquired using a Cartesian diffusion weighted spin echo sequence with 40 *b*-values ranging in magnitude from 229–7948 s/mm^2^ in both positive and negative polarities, TR/TE = 1000/31 ms, matrix size = 128 × 96, FOV = 15 × 15 mm^2^, 8 averages, total acquisition time ∼23 h. Slice thickness and number of slices varied depending on characteristics of the section. Images were reconstructed on the console itself using the vendor’s program. ADC maps were obtained using the same methods as the in vivo DWI images discussed above.

After ex vivo DWI, the tissue section was fixed in formalin and underwent standard histological processing (as described in Section 2.4). The ADC maps derived from ex vivo DWIs and the scanned H&E sections (Section 2.4) were coregistered using 2D affine transformation [21]. Image registration was implemented with MATLAB (version R2018A, MathWorks, Natick, MA, USA) functions with the ADC map as a moving image and the stained section as a fixed image. After coregistration, a grid of square ROIs (each 0.25 mm^2^) was placed on the coregistered image. For each ROI, the cell density (number of cells per unit area) was obtained in QuPath [22] (version 0.2.0 and 0.2.3) and the mean ADC value was obtained from the coregistered ADC map.

### 2.4. Histology of Tumor Tissues

Tumor tissues were harvested upon euthanasia of the mice and fixed in 10% formalin. Tissues were processed and stained by the Pathology Core of Children’s Hospital of Philadelphia. All stains were performed on formalin fixed paraffin embedded tissue sections. Staining protocols for Sirius Red, H&E and CD31 are posted on the Core’s website (https://www.research.chop.edu/pathology/tools, accessed on 16 August 2022). Stained sections were scanned at ×40 magnification using Aperio ScanScope CS2 (Leica Biosystems Imaging, CA, USA) to generate digital pictures, which were uploaded to QuPath for analyses.

We developed pixel classifiers for each stain and applied them to scanned images of the whole slide. This approach enabled unbiased assessments of the entire section as opposed to selected fields within a section. To quantitatively determine the amount of fibrosis and microvascular density as percentage of annotated area, a pixel classifier was built for Sirius Red (Appendix A) and CD31 (Appendix A) stained sections. Each classifier was developed through several rounds of annotation and training using a Random Forests algorithm. The classifiers’ performances were evaluated carefully by an experienced gastrointestinal pathologist until robust and accurate algorithms were developed.

For each section, tumor regions were defined and hand drawn by a pathologist, and % tumor area was computed. Sections stained with H&E, Sirius Red, and CD31 were used to determine cell density, % Sirius Red positive area, and % CD31 positive area, respectively.

### 2.5. Statistical Analysis

For each subject, the median value ± SD of ADC, KI, $T_{1}$, $K^{trans}$ and $v_{e}$ were obtained for tumor, spinal muscle, and phantom ROIs. Group mean ± SD of each metric was then obtained for KPC and CKS groups. To examine intra-tumor heterogeneity of the MRI metrics, SD was computed from pixels of each tumor and the mean SD was obtained for each group. Two-tailed student’s *t*-tests without assumption of equal variance with an alpha of 0.05 were conducted to evaluate the difference between parameters of KPC and CKS models.

To evaluate the effectiveness of ADC and KI in differentiating KPC from CKS tumors, univariate logistic regression models were trained using the ADC or KI as a feature and the tumor model as a label. Area under the receiver operating characteristic curve (AUROC) was used to evaluate the performance of each metric in separating KPC from CKS.

To analyze the relation between ADC and cell density obtained from co-registered ex vivo ADC maps and H&E sections, ROI data from tumors (n = 5) were combined and a least-squares linear fit was applied for cell density as a function of the reciprocal of ADC, given the inverse relationship between the two metrics. Correlation was evaluated using Spearman’s correlation coefficient, R, and a *p*-value was used to evaluate the significance of the obtained trend. A method described by Blocker et al. [23] was also implemented but produced less remarkable results due to the collapsing of ADC data in the pre-processing stages.

## 3. Results

KPC tumors appear as a single solid tumor on T2W MRI and upon gross dissection (Figure 1a,b), whereas CKS tumors appear as a lobulated mass consisting of small cystic tumors resulting in high and diffusive signals on T2W images (Figure 1g,h).

H&E sections from relatively small and large KPC tumors (Figure 1c,d, respectively) and CKS tumors (Figure 1i,j, respectively) were examined. KPC tumors feature abrupt invasive adenocarcinoma formation without intraductal growth (Figure 1e,f). In contrast, CKS tumors are initiated from multifocal epithelial proliferation within ducts of varying sizes; their growth pattern shows folding and papillary architecture resembling the IPMN in humans (Figure 1k,l). The multi-focal tumors are relatively small and surrounded by residual pancreatic acinar parenchyma (marked by * in Figure 1k), which is obliterated as the tumor grows larger (e.g., Figure 1j). The KPC adenocarcinoma formation is much smaller than the intraductal architecture of the CKS tumor (Figure 1e,f vs. Figure 1k,l). In CKS tumors, the periphery of the tumor has increased fibrosis compared to the interior of the tumor (more details in Sirius Red staining).

The in vivo DW-MRI study included 20 CKS and 44 KPC mice (reported in Figure 2 and Table 1). DW images are free of motion artifacts and distortions even at the highest *b* value (2141 mm^2^/s), leading to good quality ADC and KI maps of the tumor (Figure 2a,b). ADC values of the KPC tumors are consistent with those from human PDAC tumors (1.66 mm^2^/s in KPC vs. 1.2–1.5 mm^2^/s in human PDAC [24,25]). ADC values of CKS tumors are remarkably higher than those of KPC tumors with almost no overlap between the two models (Figure 2c and Table 1). To identify cellular architecture and histological features which underlie the distinct ADC metric between the two, histopathological analyses were performed in a group of KPC and CKS tumor specimens with a wide range of tumor sizes (Table 2). Consistent with their resemblance to histopathological characteristics of human IPMN, features corroborating to the distinctively higher ADC value of the CKS model include: (1) lower cell density as a result of multiple loosely connected colonies growing and merging, in contrast to the highly packed KPC tumor formed from a single colony, (2) the cystic nature of the tumor colonies growing in the mucinous pancreas acinar, which is absent in KPC tumors (Figure 1 and Table 2). In contrast to ADC, the KI values were significantly higher in KPC tumors and exhibit little overlap with KI values in CKS tumors(14.5 ± 3 vs. 8.86 ± 1, Figure 2d, Table 1). While the biophysical mechanism of kurtosis is not well understood, high kurtosis is usually associated with more densely packed cells or higher cellular complexity in the tumor tissues [26]. These descriptions fit will with the KPC tumor morphology although a histological marker for cellular complexity has not yet been identified.

To quantitatively evaluate the classification power of ADC and KI as metrics differentiating KPC tumors from CKS tumors, a univariate logistic regression model was trained for each metric. The model trained using ADC had an AUROC of 0.986, while the model trained using KI had an AUROC of 0.976. This suggests that both ADC and KI could serve as imaging markers to distinguish PDAC from IPMN.

To further assess the regional correlation between ADC and cell density requires anatomic coregistration of the ADC map with H&E stained slide. Here, we tested an approach using a group of KPC tumors (n = 5) by performing ex vivo DWI of a 1 mm thick slice of fresh tumor specimen, which, after ex vivo DW imaging, was embedded in the same orientation and processed for H&E staining. The ADC map and H&E slide were then coregistered using affine transformation (Figure 3a–c). A positive correlation was observed between cell density vs. the reciprocal of ADC values (R=0.33, $p=6.4\times 10^{-32}$, Figure 3d). To compare our results to other published work, we replicated Blocker et al.’s cell density vs. ADC analysis [23] using our data (Appendix A), and obtained similar results. However, we believe our method is more robust because it treats ROIs as individual samples, whereas the cited method leads to a loss of information by first grouping ROIs with similar ADC into bins, followed by fitting a curve to the mean value of these bins.

The tumor microenvironment consists of microvasculature the extracellular matrix laid down by stromal cells. Robust deposition of collagen, a major matrix component, was found surrounding individual tumor nodules of CKS tumors (Figure 4b,d,f); therefore, CKS tumors whose multiple nodules have not merged have a higher % Sirius Red positive area. In KPC tumors, collagen deposition was distributed more evenly inside and around the tumor periphery that interfaces with the pancreas (marked by * in Figure 4c). Groupwise, the CKS model exhibits twofold higher % Sirius Red than KPC (17.1±7 vs. 8.1±3, p=0.013, Table 2).

To examine the tumor microvasculature, the DCE protocol including B1 and $T_{1}$ mapping and DCE series was applied to 11 KPC and 11 CKS mice with one-to-one matched tumor size, because based on prior data, $K^{trans}$ of the KPC model is dependent on tumor size: KPC tumor size range = 160–502 mm^3^, CKS tumor size range = 178–609 mm^3^. The smaller sample size compared to the DW study is also reflective of the more recent optimization of the protocol. DCE images, tumor parametric maps, and signal time courses from ROIs are presented for two KPC and two CKS mice (Figure 5). A central region with low $K^{trans}$ surrounded by a rim of higher $K^{trans}$ is typically observed in KPC tumors with distinct tumor size and $K^{trans}$ values (Figure 5, KPC-1 vs. KPC-2). In contrast, $K^{trans}$ maps of CKS tumors do not exhibit the center/ rim pattern regardless of tumor size (Figure 5, CKS-1 vs. CKS-2). The map of extracellular extravascular volume fraction ($v_{e}$) shows higher $v_{e}$ values in locations where $K^{trans}$ values are lower in both models. Mean ± SD of median $K^{trans}$, $v_{e}$ and $T_{1}$ values from KPC and CKS groups are summarized in Table 1: KPC tumors have significantly lower $K^{trans}$ values than CKS tumors (0.188±0.01 vs. 0.253±0.05/min, p=0.034).

To evaluate the underlying mechanism of $K^{trans}$ values and their spatial distribution, we examined microvascular density using CD31 staining and classifiers (Appendix A). For KPC tumors (Figure 6a–d), the microvascular density (brown spots) is higher in the periphery of the tumors compared to the center, which is consistent with the pattern observed in $K^{trans}$ maps of KPC tumors. In KPC tumors exhibiting necrosis, strong CD31 staining is present in the viable tumor surrounding the necrotic area (Figure 6e,f), suggesting that the KPC tumor undergoes robust angiogenesis during expansion. In comparison, the microvasculature is more evenly distributed throughout the CKS tumors in both small and large tumors (Figure 6g–j), which is consistent with more uniform $K^{trans}$ maps of CKS tumors. Although spatial distribution of $K^{trans}$ and CD31 staining match with each other in both models, average % CD31 positive area in KPC tumors 2.5-fold higher than in CKS tumors (3.3 ± 0.5 vs. 1.3 ± 0.9, p=0.00075, Table 2), which appears contradictory to average $K^{trans}$ values observed in the two models (0.188±0.01 in KPC vs. 0.253±0.05/min in CKS, p=0.034, Table 1).

Intratumor heterogeneity of DW- and DCE-MRI metrics were assessed based on pixelwise parametric maps of each metric. For each metric, SD of pixel values in the tumor ROI was obtained for each mouse. Mean intratumor SD ± group SD was then calculated for CKS and KPC groups (Table 3). KPC tumors have a significantly lower intratumor variation in ADC than CKS (0.43 vs. 0.69, $p=1.0\times 10^{-8}$), consistent with KPC’s highly compact tumor bed. In contrast, KPC tumors exhibit higher intratumor variation in KI and $K^{trans}$: the former seems consistent with multiple cells packed in the tumor bed including tumor cells, cancer-associated fibroblasts, and endothelial cells, while the latter is consistent with a much less uniform distribution of microvasculature inside the tumor, as discussed above.

## 4. Discussion

In this study, DW and DCE-MRI metrics were compared between CKS, a GEM model shown to exhibit key features of human IPMN lesion, and KPC, a well-established GEM model of human PDAC. The two GEM models exhibit distinct characteristics in cellular architecture and the tumor microenvironment, including cell density, collagen content (fibrosis), microvasculature density and distribution, multifocal versus single colony growth pattern, and cystic vs. noncystic nature. We found that these characteristics underlie the biophysical properties of DW- and DCE-MRI metrics observed in the two models. Our results revealed that CKS tumors exhibit distinctly higher ADC and lower KI values than KPC tumors. Crucially, despite large numbers of animals with a wide range of tumor size included in the in vivo DW imaging study (n = 44 for KPC and n = 20 for CKS), there is little overlap of ADC (or KI) values between the two models (Figure 2c,d and Table 1). These findings are remarkable because they suggest the possibility that a DW metric (ADC or KI) could be useful as a diagnostic tool for separating benign IPMN tumors (similar to CKS) from malignant PDAC (similar to KPC).

IPMN have been recognized as important precursors to invasive pancreatic cancer [27,28,29]. Patients with IPMN, a precursor to pancreatic adenocarcinoma, are selected for surgery only if they meet certain radiographic criteria, whereas other patients are continually surveyed with repeated imaging studies [6]. While radiographic information obtained from standard CT or MRI provides estimates of the tumor size and pancreatic duct diameter, these anatomical imaging methods do not provide biological information such as cell density and microvascular density/perfusion of the tumor. Our data revealed cell density to be a primary factor underlying the tumor ADC value (Figure 3), and that differences in cell density (Table 2) are resulted from different growth patterns of KPC versus CKS tumors. Therefore, metrics from DWI and DCE-MRI can provide insight related to the tumor’s degree of malignancy or features of the tumor microenvironment such as microvasculature. The stratification based on DW-MRI metrics demonstrated in GEM models may be translated to clinical use, where an IPMN lesion that is progressing to invasive PDAC could be suspected by a reduced ADC and an increased KI value relative to a threshold value. Such clinical translation would certainly require extensive validations but is facilitated by the fact that DW-MRI is widely used clinically in oncology. Prior studies of DW-MRI in patients with IPMN found ADC values for low grade IPMN is higher than those for high-grade or invasive IPMN [30].

Mouse models for IPMN are lacking. Besides the CKS studied here, another model based on GNAS and Kras mutations is reported to exhibit phenotypes of IPMN-like growth patterns and cytology similar to those of CKS [31]. Therefore, the CKS model gives a readily available model with multifocal intraductal growth amenable to study at differing stages and sizes of progression. Smad4 deletion in CKS also enables studies of Smad4 and the downstream TGF*β* pathway in tumor progression and microenvironment [32]. The higher level of fibrosis featured by extracellular collagen deposition is shown in CKS tumors (Figure 4). This is consistent with Smad4 deletion, which activates the TGF*β* pathway downstream, leading to matrix protein deposition [4,33]. We expect the higher extracellular matrix collagen levels in the CKS model to be detected by magnetization transfer techniques, which has been applied to evaluate the extent of fibrosis in tumors and organs such as the liver and kidneys [34,35,36].

$K^{trans}$ maps derived from DCE-MRI revealed different patterns of perfusion and/or vascular permeability in KPC versus CKS tumors (Figure 5); these patterns are consistent with their CD31 staining pattern (Figure 6). Compared to CKS tumors, KPC tumors grow significantly more new capillaries (Table 2) but have significantly lower $K^{trans}$ (Table 1). This apparently paradoxical finding could be explained by the tumor’s architecture and microenvironment. Besides a highly compact tumor bed, deposition of collagen and hyaluronan (a glycosaminoglycan) in the extracellular matrix leads to high interstitial fluid pressure in KPC tumors [37,38] because both collagen and hyaluronan absorb large numbers of water molecules. Consequently, high interstitial fluid pressure could collapse or shut down vascular perfusion and/or permeability, leading to lower $K^{trans}$ values despite abundant neovasculature formation.

Quantitative DW- and DCE-MRI’s sensitivity to biophysical features such as diffusion, permeability and perfusion has led to development of several computational models of tumor growth, microvascular characterization, and hemodynamics [39], and response to radiation therapy [40]. The modeling approach allows biophysical insights at cellular level and is akin to computational models used in drug discovery to assess molecular interactions between drug molecule and its biological target glucose transporter or HIV [41,42].

## 5. Conclusions

In this study, we found that apparent diffusion coefficient (ADC) and kurtosis index (KI) are two quantitative diffusion-weighted imaging metrics that can reliably differentiate between two GEM models: CKS, which models for intraductal papillary mucinous neoplasms (IPMN), a premalignant disease, and KPC, a model for pancreatic ductal adenocarcinoma. ADC was shown to inversely correlate with cell density, both locally within individual tumors and when comparing the two tumor models. $K^{trans}$, a DCE-MRI metric, mapped the pattern of microvasculature within tumors. These results suggest metrics derived from DW-MRI might be used to identify IPMN lesions that are progressing to malignancy and as such, should be investigated clinically. 

## Figures and Tables

**Figure 1 cancers-14-04017-f001:**
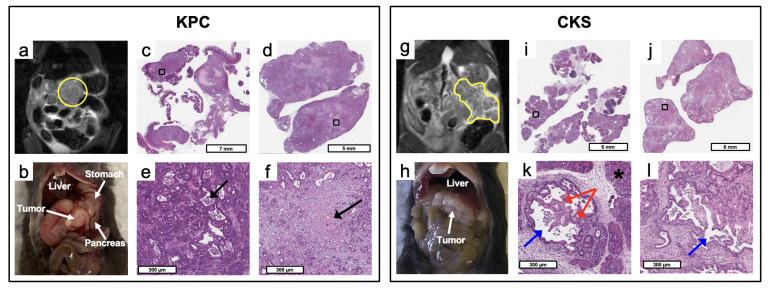
MRI, gross dissection, and pathological features of KPC and CKS tumors. (**a**) T2W coronal MRI view of a KPC mouse, (**b**) gross dissection, (**c**) H&E stained micrograph of a relatively small and (**d**) large KPC tumor specimen. (**e**) is the magnified region labeled in (**c**), showing the glandular structure (black arrow) formed by surrounding adenocarcinoma cells. (**f**) is the magnified necrotic tumor region (black arrow) labeled in (**d**). (**g**) T2W coronal MRI view of a CKS mouse, (**h**) gross dissection, (**i**) H&E stained micrograph of a relatively small and (**j**) large CKS tumor specimen. (**k**) is the magnified labeled region labeled in (**i**), revealing folding and papillary architecture formed by tumor cells (red arrow) growing inside the lumen (blue arrow) of an expanded pancreatic duct and residual pancreas (*). (**l**) is the magnified labeled region in (**j**), showing the lumen (blue arrow) of a duct expanded by neoplastic ductal epithelial cells.

**Figure 2 cancers-14-04017-f002:**
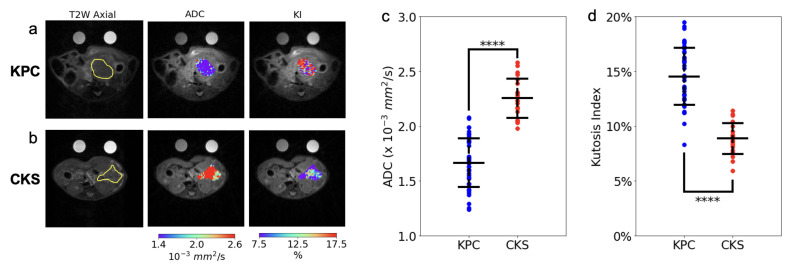
Comparison of quantitative in vivo DW-MRI markers between KPC and CKS tumors. (**a**,**b**) display a T2W axial image and tumor ADC (color) and KI (color) map overlays, for a KPC and CKS tumor, respectively. (**c**,**d**) present ADC and KI values from individual KPC (n = 44) and CKS tumors (n = 20) along with the group mean and standard deviation. *p*-values are from two-tailed *t*-tests. ****: *p* < $10^{-10}$.

**Figure 3 cancers-14-04017-f003:**
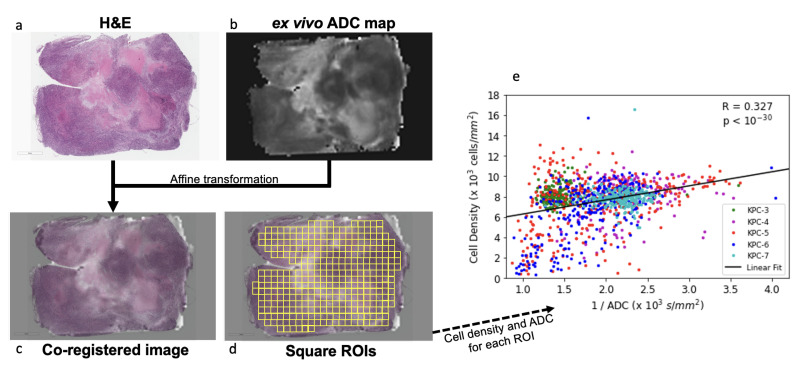
Correlation between ADC and cell density in KPC model. (**a**) H&E stained section and (**b**) ex vivo ADC map were transformed to the same space via affine transformation to produce a (**c**) coregistered image. (**d**) Square ROIs were placed covering the entire section show a negative correlation (**e**) between ADC and cell density (R=−0.38, $p=1.6\times 10^{-43}$).

**Figure 4 cancers-14-04017-f004:**
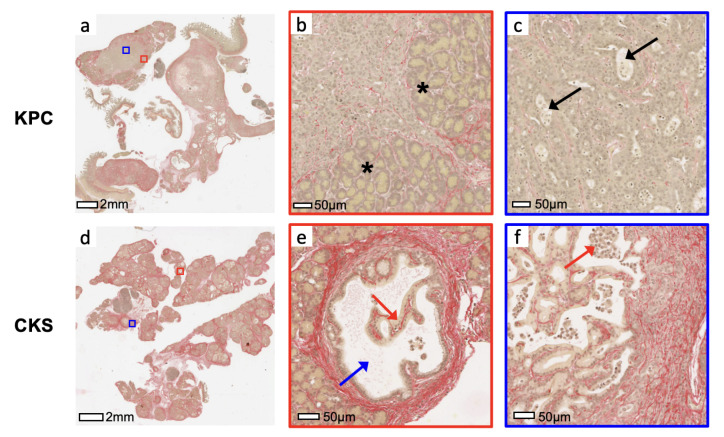
Sirius red staining of a KPC and CKS tumor. Sirius red staining of KPC tumor specimen (**a**) reveals collagen (stained red) deposition around tumor cells and residual pancreatic glands (* in **b**). Glandular structures (arrows in **c**) are formed by surrounding tumor cells, with necrotic tumor cells are seen inside the glandular space. In CKS tumor specimen (**d**), strong depositions of collagen fibers are found surrounding the pancreatic ducts (**e**), with tumor cells (red arrow) growing inside the lumen (blue arrow) folding and forming a papillary architecture. Magnified region (**f**) shows collagen depositions are also found surrounding and tumor nodules and the papillary formation of necrotic tumor cells inside expanded ducts (red arrow).

**Figure 5 cancers-14-04017-f005:**
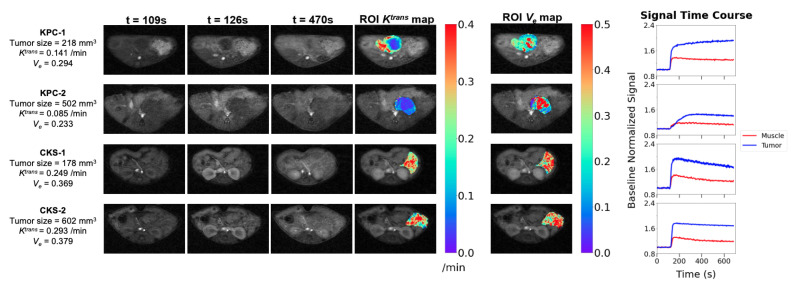
Comparison of quantitative DCE-MRI features and parametric maps between KPC and CKS tumors. For 2 KPC and 2 CKS mice, contrast enhanced images before, shortly after, and minutes after contrast agent bolus are shown, as well as resulting $K^{trans}$ and $v_{e}$ maps of the tumor ROI and the signal time-course for the full tumor and spinal muscle (reference region) ROIs. *t* marks the time after initiation of DCE acquisition. Contrast media is injected at 120 s. Images are cropped. Refer to Appendix A for an example of ROI placed for spinal muscle.

**Figure 6 cancers-14-04017-f006:**
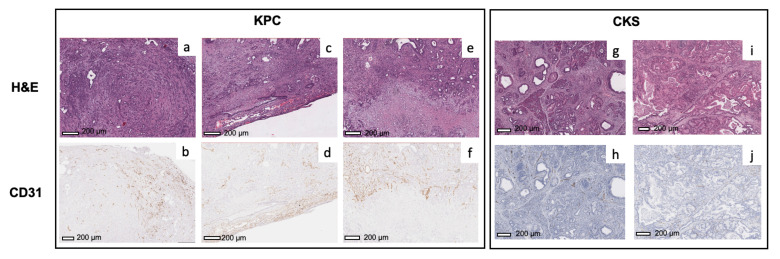
CD31 staining of KPC and CKS tumor. CD31 stained micrographs are shown for two KPC tumor specimens (**b**,**d**,**f**) with the corresponding H&E micrographs from adjacent sections (**a**,**c**,**e**). CD31 stained micrographs are shown for two CKS tumor specimens (**h**,**j**) with the corresponding H&E micrographs (**g**,**i**). The blue tint in CKS slides from hematoxylin counter staining does not interfere with quantification of microvasculature density by the classifier.

**Table 1 cancers-14-04017-t001:** Quantitative DW- and DCE-MRI metrics ^a,b,c^.

	ADC ($10^{-3}$ mm^2^/s)	KI (%)	$T_{1}$ (s)	$K^{\mathrm{trans}}$ (/min)	$v_{e}$
KPC	1.66±0.2 **** (n = 44)	14.5±3 **** (n = 44)	2.59±0.3 ** (n = 11)	0.188±0.07 * (n = 11)	0.303±0.04 (n = 11)
CKS	2.25±0.2 (n = 20)	8.86±1 (n = 20)	2.24±0.2 (n = 11)	0.253±0.05 (n = 11)	0.346±0.06 (n = 11)
Phantom 1	2.46±0.1	1.93±1	3.37±0.8	NR	NR
Phantom 2	1.39±0.1	0.18±0.4	2.80±0.9	NR	NR

a: Data presented as mean ± SD; b: *p*-values were obtained from two-tailed student *t*-test with alpha set at 0.05. c: DCE metrics (*K^trans^* and *v_e_*) were obtained from 11 KPC and 11 CKS mice with one-to-one matches tumor size ranging from approximately 100–500 mm^3^ measured on T2W MRI. *: *p* < 0.05; **: *p* < 0.01; ****: *p* < 10^−10^; NR: Not relevant.

**Table 2 cancers-14-04017-t002:** Summary of immunohistochemistry analyses of KPC and CKS tumors ^a,b^.

	Cell Density (Cells /mm^2^)	Sirius Red (%)	CD31 (%)	Necrotic Tumor (/min)
KPC	7100±300	8.1±3	3.3±0.5	3 out of 5
CKS	6120±800	17.1±7	1.3±0.9	0 out of 7
*p*-value ^c^	**0.016** *	**0.013** *	**0.00075** **	N/A

a: Data presented as mean ± SD; b: *p*-values were obtained from two-tailed student *t*-test with alpha set at 0.05. c: On micrographs of tumor specimens, % tumor area (dissected from the entire tissue area) ranges from 29–89% (mean = 62%) for KPC, and 8–100% (mean 51%) for CKS. Parameters reported in the Table are derived from the entire section, since it is not possible to make such differentiation on MR images. *: *p* < 0.05; **: *p* < 0.01; when comparing KPC vs. CKS groups.

**Table 3 cancers-14-04017-t003:** Intratumor standard deviations of DW- and DCE-MRI metrics ^a,b^.

	SD of ADC ($10^{-3}$ mm^2^/s)	SD of KI (%)	SD of $T_{1}$ (s)	SD of $K^{\mathrm{trans}}$ (/min)	SD of $v_{e}$
KPC	0.433±0.01 (n = 44)	6.66±2 (n = 44)	0.538±0.4 (n = 11)	0.160±0.04 (n = 11)	0.172±0.05 (n = 11)
CKS	0.689±0.1 (n = 20)	5.36±1 (n = 20)	0.535±0.01 (n = 11)	0.122±0.02 (n = 11)	0.145±0.03 (n = 11)
*p*-value	$1.0\times 10^{-8}$ ***	0.0013 **	0.980	0.019 *	0.18

a: Data presented as mean ± SD; b: *p*-values were obtained from two-tailed student *t*-test with alpha set at 0.05. *: *p* < 0.05, **: *p* < 0.01, ***: *p* < 0.001 when comparing KPC vs. CKS groups.

## Data Availability

Data supporting the results of this study will be deposited on the program’s GitHub repository https://pennpancreaticcancerimagingresource.github.io/data.html (accessed on 16 August 2022) upon publication of the paper.

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
