# Peer review of "DWI Metrics Differentiating Benign Intraductal Papillary Mucinous Neoplasms from Invasive Pancreatic Cancer: A Study in GEM Models"

_cancers, 2022, doi:10.3390/cancers14164017_

Round 1

Reviewer 1 Report

Dear Authors,

This is an interesting, well conducted and written research article.

Author Response

Point 1: This is an interesting, well conducted and written research article.

Response 1: Thank you for your comments!

Reviewer 2 Report

In this submission to Cancer, the authors compare autochthonous tumors using quantitative imaging metrics from diffusion-weighted (DW) and dynamic contrast enhanced (DCE)-MRI in reference to quantitative histological metrics including cell density, fibrosis, and microvasculature density. The authors' results revealed distinct DW-MRI 

metrics between the KPC vs. CKS model (mimicking human PDAC vs. IPMN lesion). Specifically, the authors find that the apparent diffusion coefficient (ADC) of CKS tumors is significantly higher than that of KPC, with little overlap despite intratumor and inter-tumor variability. The authors conclude that their studies in GEM models demonstrate the potential utility of diffusion-weighted MRI metrics for distinguishing pancreatic cancer from benign pancreatic cysts such as IPMN.

I consider this manuscript to be of interest to readers of this journal. As such, I am supportive of publication with a minor, but required revision.  Specifically, there has been much work using computational approaches to understand biological systems at the atomistic level, which can be used to complement experimental work in cancer, which should be mentioned:

Journal of Molecular Structure 2021, 1227, 129511

Journal of Chemical Theory and Computation, 2019, 15, 2807-2815

In particular, these prior works have shown that computational approaches such as DFTB can provide additional details to experiment and give insight into atomistic/mechanisms in processes such as cancer. As such, these prior works should probably be mentioned in the next revision. With this revision, I would be receptive towards a re-review of this manuscript.

Author Response

Point 1: 

In this submission to Cancer, the authors compare autochthonous tumors using quantitative imaging metrics from diffusion-weighted (DW) and dynamic contrast enhanced (DCE)-MRI in reference to quantitative histological metrics including cell density, fibrosis, and microvasculature density. The authors' results revealed distinct DW-MRI metrics between the KPC vs. CKS model (mimicking human PDAC vs. IPMN lesion). Specifically, the authors find that the apparent diffusion coefficient (ADC) of CKS tumors is significantly higher than that of KPC, with little overlap despite intratumor and inter-tumor variability. The authors conclude that their studies in GEM models demonstrate the potential utility of diffusion-weighted MRI metrics for distinguishing pancreatic cancer from benign pancreatic cysts such as IPMN.

I consider this manuscript to be of interest to readers of this journal. As such, I am supportive of publication with a minor, but required revision.  Specifically, there has been much work using computational approaches to understand biological systems at the atomistic level, which can be used to complement experimental work in cancer, which should be mentioned:

Journal of Molecular Structure 2021, 1227, 129511

Journal of Chemical Theory and Computation, 2019, 15, 2807-2815

In particular, these prior works have shown that computational approaches such as DFTB can provide additional details to experiment and give insight into atomistic/mechanisms in processes such as cancer. As such, these prior works should probably be mentioned in the next revision. With this revision, I would be receptive towards a re-review of this manuscript.

Response 1:

Thank you for your suggestion. We agree that computational approaches to understanding biological systems, particularly at the atomistic level are an exciting area of current research and can be applied to drug discovery and development in oncology. Accordingly, we have added a paragraph on the potential of these approaches in the Discussion (see attached, lines 399-405).

Reviewer 3 Report

The authors proposed a promising method for differential diagnosis between non-invasive and invasive IPMN based on DW-MRI metrics which provides insights into the tumor microenvironment. The experiment design as well as statistic analysis are good. However, I have some concerns which are listed below.

1. At first glance, the title is a little bit confusing and can be misleading. Under most conditions, it’s not difficult to differentiate between typical IPMN and traditional PDAC based on the imaging results (one is cystic neoplasm and the other is hypovascular solid tumor). After reading the paper, I realized that the authors actually intended to find more sensitive and quantitative indicators depending on MRI metrics to differentiate between non-invasive and invasive IPMN, which can be used as high-risk factors during surveillance. The malignant transformation or progression from IPMN to malignancy is referred to as “IPMN with an invasive carcinoma” or “intraductal papillary mucinous carcinoma” (IPMC). IPMC could be further classified into tubular type and colloid type. Although the tubular subtype shares similar morphological characteristics with conventional PDAC, there may be a distinction between PDAC derived from IPMN (IPMC) and conventional PDAC with respect to imaging features. The traditional high-risk stigmata for IPMN malignant transformation include MPD dilatation 10 mm, enhanced solid component, cyst 3 cm, etc. KPC model is widely used as the animal model for conventional PDAC.  Does this model serve well for the research purpose?  The authors should discuss this more.

2. GEM is a big selling point of this paper. There are very limited studies reporting IPMN animal models, and I failed to find other papers reporting the CKS model except for the one cited in the manuscript. The authors should introduce more about the characteristics of the CKS model and/or cite more references for this model.

3. IPMN can be clinically classified into three types: main-duct, branch-duct, and mixed type. Surgical strategy is complicated and dependent on classification1,2,3. In general, for main-duct and mixed type IPMN, surgical decisions are more radical compared to branch-duct type. Most surgeons would recommend surgical resection by the time of their diagnosis. So, in line 33, “IPMN is considered a premalignant disease, therefore, surveillance of tumor progression rather than surgical excision is recommended for the majority of IPMN patients.”  need to be changed to a more accurate expression.

1. Nakamura, Masafumi, et al. "Comparison of guidelines for intraductal papillary mucinous neoplasm: What is the next step beyond the current guidelines?." Annals of Gastroenterological Surgery 1.2 (2017): 90-98.

2. Tanaka, Masao, et al. "International consensus guidelines 2012 for the management of IPMN and MCN of the pancreas." Pancreatology 12.3 (2012): 183-197.

3. Tanaka, Masao, et al. "Revisions of international consensus Fukuoka guidelines for the management of IPMN of the pancreas." Pancreatology 17.5 (2017): 738-753.

Author Response

Point 1: 

At first glance, the title is a little bit confusing and can be misleading. Under most conditions, it’s not difficult to differentiate between typical IPMN and traditional PDAC based on the imaging results (one is cystic neoplasm and the other is hypovascular solid tumor). After reading the paper, I realized that the authors actually intended to find more sensitive and quantitative indicators depending on MRI metrics to differentiate between non-invasive and invasive IPMN, which can be used as high-risk factors during surveillance. The malignant transformation or progression from IPMN to malignancy is referred to as “IPMN with an invasive carcinoma” or “intraductal papillary mucinous carcinoma” (IPMC). IPMC could be further classified into tubular type and colloid type. Although the tubular subtype shares similar morphological characteristics with conventional PDAC, there may be a distinction between PDAC derived from IPMN (IPMC) and conventional PDAC with respect to imaging features. The traditional high-risk stigmata for IPMN malignant transformation include MPD dilatation ≥10 mm, enhanced solid component, cyst ≥3 cm, etc. KPC model is widely used as the animal model for conventional PDAC.  Does this model serve well for the research purpose?  The authors should discuss this more.

Response 1: 

We agree that the title did not precisely describe the potential use case for our findings. Accordingly, we have revised the title of the article (see attached) to: "DWI Metrics Differentiating Benign Intraductal Papillary Mucinous Neoplasms from Invasive Pancreatic Cancer: A study in GEM Models".

Regarding the use of the KPC model as a model for IPMC, we agree that the distinction between conventional PDAC and some subtypes of IPMC exist, but as the reviewer mentioned, the more prevalent and lethal type of IPMC  presents similar histopathological features to conventional PDAC . Therefore, we believe the KPC model is suitable as a model of more invasive carcinoma arising from IPMN but may not fully capture the continuum of invasiveness in a progressing IPMN (see our revision in lines 55-61).  

Point 2:

GEM is a big selling point of this paper. There are very limited studies reporting IPMN animal models, and I failed to find other papers reporting the CKS model except for the one cited in the manuscript. The authors should introduce more about the characteristics of the CKS model and/or cite more references for this model.

Response 2:

We have added additional information on the CKS mouse model as the reviewer suggested in two places: Section 2.1 of the Methods section (lines 90-95) and in the discussion (lines 376-381).

Point 3:

IPMN can be clinically classified into three types: main-duct, branch-duct, and mixed type. Surgical strategy is complicated and dependent on classification[1,2,3]. In general, for main-duct and mixed type IPMN, surgical decisions are more radical compared to branch-duct type. Most surgeons would recommend surgical resection by the time of their diagnosis. So, in line 33, “IPMN is considered a premalignant disease, therefore, surveillance of tumor progression rather than surgical excision is recommended for the majority of IPMN patients.”  need to be changed to a more accurate expression.

  1. Nakamura, Masafumi, et al. "Comparison of guidelines for intraductal papillary mucinous neoplasm: What is the next step beyond the current guidelines?." Annals of Gastroenterological Surgery 1.2 (2017): 90-98.
  2. Tanaka, Masao, et al. "International consensus guidelines 2012 for the management of IPMN and MCN of the pancreas." Pancreatology 12.3 (2012): 183-197.
  3. Tanaka, Masao, et al. "Revisions of international consensus Fukuoka guidelines for the management of IPMN of the pancreas." Pancreatology 17.5 (2017): 738-753.

Response 3:

We have changed our statement in line 33 (now line 39) to be more precise, addressing certain features of the IPMN that are considered in the decision to resect surgically or not (lines 39-46). We have also added a sentence to the same effect in the Discussion (lines 361-366).
